# Structural basis of ligand binding modes at the human formyl peptide receptor 2

Tong Chen[1,2,3,8], Muya Xiong [1,3,8], Xin Zong[1,2,3], Yunjun Ge[4], Hui Zhang[1,2,3], Mu Wang[1,5], Gye Won Han[6], Cuiying Yi[1], Limin Ma[2], Richard D. Ye [7], Yechun Xu [1,3✉], Qiang Zhao[2,3✉] & Beili Wu [1,3,5✉]

The human formyl peptide receptor 2 (FPR2) plays a crucial role in host defense and inflammation, and has been considered as a drug target for chronic inflammatory diseases. A variety of peptides with different structures and origins have been characterized as FPR2 ligands. However, the ligand-binding modes of FPR2 remain elusive, thereby limiting the development of potential drugs. Here we report the crystal structure of FPR2 bound to the potent peptide agonist WKYMVm at 2.8 Å resolution. The structure adopts an active conformation and exhibits a deep ligand-binding pocket. Combined with mutagenesis, ligand binding and signaling studies, key interactions between the agonist and FPR2 that govern ligand recognition and receptor activation are identified. Furthermore, molecular docking and functional assays reveal key factors that may define binding affinity and agonist potency of formyl peptides. These findings deepen our understanding about ligand recognition and selectivity mechanisms of the formyl peptide receptor family.

[1] CAS Key Laboratory of Receptor Research, Shanghai Institute of Materia Medica, Chinese Academy of Sciences, 555 Zuchongzhi Road, 201203 Pudong, Shanghai, China. [2] State Key Laboratory of Drug Research, Shanghai Institute of Materia Medica, Chinese Academy of Sciences, 555 Zuchongzhi Road, 201203 Pudong, Shanghai, China. [3] University of Chinese Academy of Sciences, No.19A Yuquan Road, 100049 Beijing, China. [4] Institute of Chinese Medical Sciences, University of Macau, 999078 Macau Special Administrative Region, China. [5] School of Life Science and Technology, ShanghaiTech University, 393 Hua Xia Zhong Road, 201210 Shanghai, China. [6] Department of Chemistry, Bridge Institute, Michelson Center for Convergent Bioscience, University of Southern California, 1002 West Childs Way, Los Angeles, CA 90089, USA. [7] School of Life and Health Sciences, The Chinese University of Hong Kong, 518172 Shenzhen, China. [8]These authors contributed equally: Tong Chen, Muya Xiong. ✉email: ycxu@simm.ac.cn; zhaoq@simm.ac.cn; beiliwu@simm.ac.cn

Three formyl peptide receptors (FPR1, FPR2, and FPR3), which belong to the GPCR superfamily, participate in many physiological processes in humans[1]. These receptors bind a vast array of structurally diverse agonists, including N-formyl peptides from microbes and mitochondria, and non-formyl peptides of microbial and host origins[1,2]. FPR2, also known as the lipoxin A4 receptor (LXA4R, ALX), plays important roles in chemotaxis, cell proliferation, wound healing, migration, and vessel growth, and is involved in the pathogenesis of chronic inflammatory diseases such as rheumatoid arthritis, colitis, Alzheimer's disease, systemic amyloidosis and atherosclerosis[2,3]. Although FPR1 and FPR2 share 69% sequence identity, FPR2 displays low affinity binding to the prototypic formyl peptide, N-formyl-Met-Leu-Phe (fMLF) and many potent formyl peptide agonists for FPR1[4]. However, FPR2 can recognize a wider array of ligands with different structures and functions, which include not only bacterially derived formyl peptides but also non-formyl peptides, lipid mediators such as lipoxin A4 (LXA4), small molecules and proteins[2,5,6], making this receptor one of the most promiscuous GPCRs characterized to date. How FPR2 is able to recognize and bind these ligands and transduce both proinflammatory and anti-inflammatory signals remains a mystery. Some FPR2 ligands have shown therapeutic potential for the treatment of inflammation, diabetic wounds and Alzheimer's disease[7–9]. However, the lack of a three-dimensional structure of FPR2 has hampered the understanding of the potential therapeutic mechanism as well as their clinical applications. Trp-Lys-Tyr-Met-Val-D-Met-NH$_2$ (WKYMVm), a highly potent FPR2 agonist isolated through a library screening of synthetic peptides, showed therapeutic effects on cutaneous wound healing, coronary artery stenosis and ischemic neovascularization, and has been considered as a promising drug candidate[10]. To provide molecular details of ligand recognition by FPR2 and better understand the ligand-binding behavior of different formyl peptide receptors, we determined the crystal structure of FPR2 in complex with WKYMVm. This structure, together with mutagenesis, ligand binding, receptor signaling and molecular docking studies, reveals critical receptor-ligand interactions that define recognition of various ligands by FPR2 and identifies key factors that may govern receptor signaling.

## Results

### The FPR2-WKYMVm structure adopts an active conformation.

To facilitate crystal packing, the N-terminal residues M1-E2 of FPR2 were replaced with a thermostable apocytochrome b562RIL (bRIL)[11] fusion protein and five residues at C terminus were truncated. A single mutation S211$^{5.48}$L (superscript indicates residue numbering using the Ballesteros Weinstein nomenclature[12]) was introduced to further improve protein quality. It was designed by switching the hydrophilic residue to a hydrophobic counterpart presented in several class A GPCRs with known structures, including chemokine receptors CXCR4 and CCR5, C5α receptor, μ-opioid receptor (μOR) and prostanoid receptor DP$_2$, which share high sequence similarity with FPR2 (35–45%)[13–17]. This mutation may introduce extra hydrophobic interactions with its neighboring residues on the external surface of the receptor, aiming for better protein stability. Functional assays indicate that the above modifications have little effect on ligand binding and receptor activation (Supplementary Tables 1, 2). The modified FPR2 protein was co-purified and co-crystallized with the peptide agonist WKYMVm. The FPR2-WKYMVm complex structure was determined at 2.8 Å resolution (Supplementary Table 3).

The FPR2 structure exhibits a canonical seven-transmembrane helical bundle conformation (helices I–VII) (Fig. 1). Some conserved GPCR structural features are observed in the extracellular region of the receptor, including a disulfide bridge connecting helix III and the second extracellular loop (ECL2) and a β-hairpin conformation of ECL2, which is shared by other solved peptide class A GPCR structures. The extracellular region, mainly including the N terminus, the first extracellular loop (ECL1) and ECL2, forms a "lid" conformation that stacks on top of the ligand-binding pocket of FPR2 (Fig. 1b). However, the structure does not rule out the possibility that the conformation of the receptor N terminus was affected by the N-terminal bRIL fusion protein, which is involved in mediating crystal packing (Supplementary Fig. 1).

The WKYMVm-bound FPR2 structure exhibits an outward shift of helix VI that is not seen in the inactive μOR structure[18]. This movement of helix VI, however, is similar to that observed in the active μOR structure[19] (Fig. 2a), suggesting that the FPR2 structure adopts an active conformation. The "ionic lock"

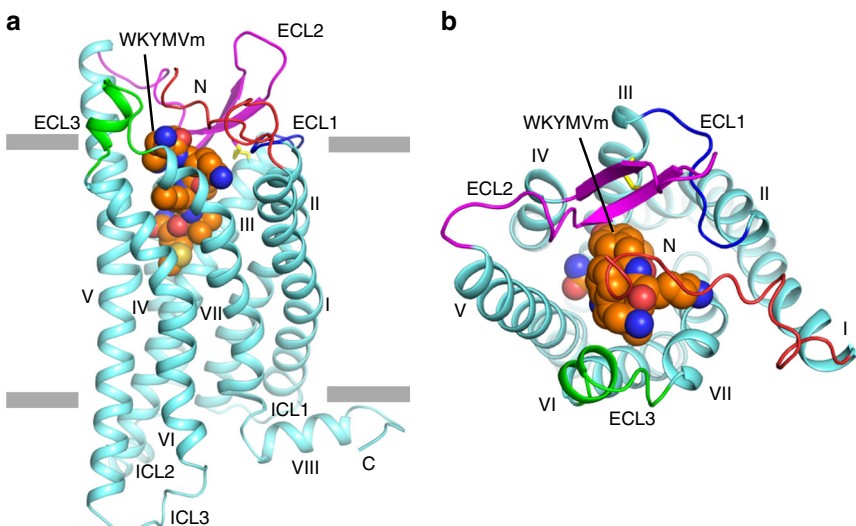

**Fig. 1 Overall structure of the FPR2-WKYMVm complex. a** Side view of the FPR2-WKYMVm structure. The receptor is shown in cartoon representation and colored cyan. The N terminus and the extracellular loops, ECL1, ECL2 and ECL3, of the receptor are colored red, blue, magenta, and green, respectively. The peptide WKYMVm is shown as spheres with carbons in orange. The disulfide bond is shown as yellow sticks. The membrane boundaries are indicated by gray blocks. **b** Extracellular view of the FPR2-WKYMVm structure.

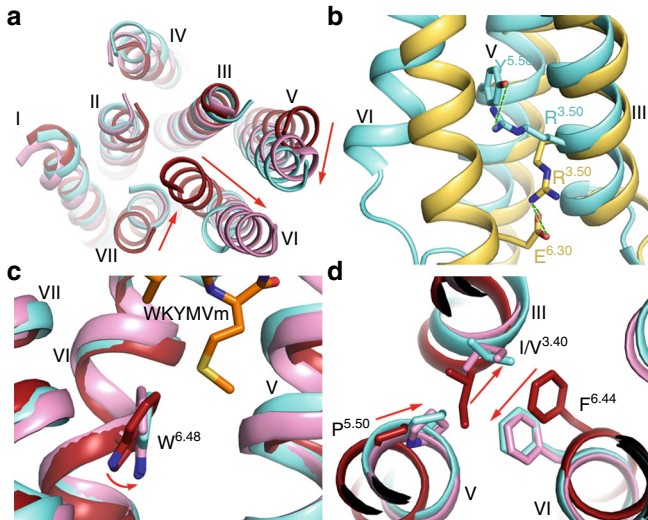

**Fig. 2 Active conformation of the FPR2-WKYMVm complex. a** The movement of helix VI. The transmembrane helical bundles in the structures of FPR2-WKYMVm (cyan), μOR-BU72 (pink, PDB code: 5C1M) and μOR-naloxone (dark red, PDB code: 4DKL) are shown in cartoon representation at an intracellular view. The red arrows indicate the movements of helices V, VI, and VII in the FPR2-WKYMVm and μOR-BU72 (active) structures relative to the μOR-naloxone (inactive) structure. **b** The hydrogen bond interaction between $R^{3.50}$ and $Y^{5.58}$ in FPR2. The FPR2 residues $R^{3.50}$ and $Y^{5.58}$ are shown as cyan sticks. The inactive $A_1AR$-DU172 structure (PDB code: 5UEN) is shown in yellow cartoon representation. The $A_1AR$ residues $R^{3.50}$ and $E^{6.30}$ that form a salt bridge are shown as yellow sticks. The polar interactions are shown as green dashed lines. **c** The conformational change of $W^{6.48}$. The structures of FPR2-WKYMVm (cyan), μOR-BU72 (pink), and μOR-naloxone (dark red) are shown as cartoons. The residue $W^{6.48}$ in the three structures is shown as sticks. The red arrow indicates the movement of $W^{6.48}$ side chain in the FPR2-WKYMVm and μOR-BU72 (active) structures relative to the μOR-naloxone (inactive) structure. **d** The conformational change of the $P^{5.50}I/V^{3.40}F^{6.44}$ motif. The residues at positions 3.40, 5.50, and 6.44 in the structures of FPR2-WKYMVm (cyan), μOR-BU72 (pink), and μOR-naloxone (dark red) are shown as sticks. The red arrows indicate the relocations of these residues in the FPR2-WKYMVm and μOR-BU72 (active) structures relative to the μOR-naloxone (inactive) structure.

between the conserved class A GPCR residue $R^{3.50}$ in the D/ER$^{3.50}$Y motif and D/E$^{6.30}$ in helix VI, which is observed in some inactive GPCR structures and has been suggested to stabilize the receptor in an inactive state[20,21], breaks in the FPR2-WKYMVm structure due to the outward shift of helix VI. Instead, the residue R123$^{3.50}$ makes a hydrogen-bond interaction with Y221$^{5.58}$ (Fig. 2b). This interaction was also observed in other active GPCR structures and has been suggested to stablilize the receptor helix V in an active orientation[19,22–26]. Moreover, the highly conserved residue $W^{6.48}$, which represents the "toggle switch"[27], and the $P^{5.50}I/V^{3.40}F^{6.44}$ motif display rotamer conformational changes in the WKYMVm-bound FPR2 structure relative to the inactive μOR structure, and adopt similar conformations to those in the active μOR structure (Fig. 2c, d). These two "microswitches" have been reported to be involved in receptor conformational rearrangement, which is required for GPCR activation[16,22,28]. The above structural features demonstrate that the FPR2-WKYMVm structure is in an active conformation.

**Binding mode of FPR2 to WKYMVm.** The peptide agonist WKYMVm binds to FPR2 in a pocket bordered by N terminus,

ECL1, ECL2, ECL3, and helices III, V, VI, and VII of the receptor (Figs. 1b, 3a, b, and Supplementary Fig. 2). The peptide penetrates into the binding cavity with its C terminus occupying a deeper site within the receptor transmembrane helical bundle compared to the binding sites in other known peptide-bound GPCR structures (Fig. 3c and Supplementary Fig. 3), while the N terminus of the peptide approaches the extracellular surface of the receptor and forms contacts with the extracellular loops. A close inspection of the ligand-binding pocket has found two hydrophobic clusters that play critical roles in ligand recognition and receptor activation. The residues F5, L164, F178, F180, L198$^{5.35}$, L268, and M271 in N terminus, ECL2, ECL3, and helix V together build a groove to accommodate the N-terminal aromatic residues W1 and Y3 of the peptide ligand (Fig. 3d). At the bottom of the ligand-binding pocket, another hydrophobic cluster, including V105$^{3.32}$, L109$^{3.36}$, F110$^{3.37}$, V113$^{3.40}$, W254$^{6.48}$, F257$^{6.51}$, and F292$^{7.43}$ in helices III, VI, and VII, forms close contacts with the C-terminal residues V5 and m6 (D-Met6) of the peptide agonist (Fig. 3e). Previous investigation of structure–activity relationship of FPR2 hexapeptide ligands revealed that the replacement of Y3, P5, or M6 in the peptide MKYMPM-NH$_2$, an FPR2 agonist related to WKYMVm with two substitutions at positions 1 and 5 and the replacement of D-Met with L-Met at C terminus, resulted in a dramatic loss of phosphoinositide hydrolysis stimulatory activity, indicating that these three residues are essential for agonist activity of the hexapeptide[29]. This finding suggests that the interactions (mainly hydrophobic) between the receptor and peptide residues at positions 3, 5, and 6 observed in the FPR2-WKYMVm structure play critical roles in mediating agonistic activity and signaling through the receptor. The importance of these hydrophobic clusters in ligand recognition was reflected in our ligand-binding assay, showing that the mutations V105$^{3.32}$F, L109$^{3.36}$A, F110$^{3.37}$A, V113$^{3.40}$A, L164W, F178A, F180A, W254$^{6.48}$A, F257$^{6.51}$A, and M271A abolish the binding of FPR2 to WK(FITC)YMVm (Fig. 4a, b and Supplementary Table 1). These mutations were further tested by an inositol phosphate (IP) accumulation assay using a chimeric Gα protein Gα$_{\Delta 6qi4myr}$[30], which facilitates the coupling of G$_i$-bound receptor to the phospholipase C signaling pathway. The results show that the alanine replacements of V105$^{3.32}$, L109$^{3.36}$, V113$^{3.40}$, L164, F178, F180, L198$^{5.35}$, W254$^{6.48}$, and F257$^{6.51}$ reduce the EC$_{50}$ of WKYMVm-induced IP production by over 65-fold (Fig. 4e, f and Supplementary Table 2). The effect of these mutations on ligand binding and receptor signaling could be explained by direct disruption of the receptor-ligand interaction, destabilization of ligand-binding pocket conformation, and/or impairment of global conformational rearrangement required for receptor activation. In contrast to the substantial effect of most of the residues within the two hydrophobic clusters, the alanine mutation of the N-terminal residue F5 displays little effect in both assays (Supplementary Tables 1, 2), suggesting that the interaction between the receptor N terminus and the peptide agonist is either not important for ligand recognition or introduced by crystal packing (Supplementary Fig. 1).

Two polar residues D$^{3.33}$ and R$^{5.38}$ in FPR1 have been suggested to form hydrogen-bond interactions with the N-terminal formyl group of fMLF[31]. In the FPR2-WKYMVm structure, the corresponding residues in FPR2 establish a hydrogen-bond network with the side chain hydroxyl of Y3, the main chain carbonyl of M4, the main chain nitrogen of m6 and the C-terminal amide group in the peptide agonist (Fig. 3f), thereby greatly contributing to the receptor-ligand interaction and stabilizing the peptide in a conformation favoring its binding to the receptor. Consistent with the importance of these two polar residues, the mutations D106$^{3.33}$A and R201$^{5.38}$A significantly

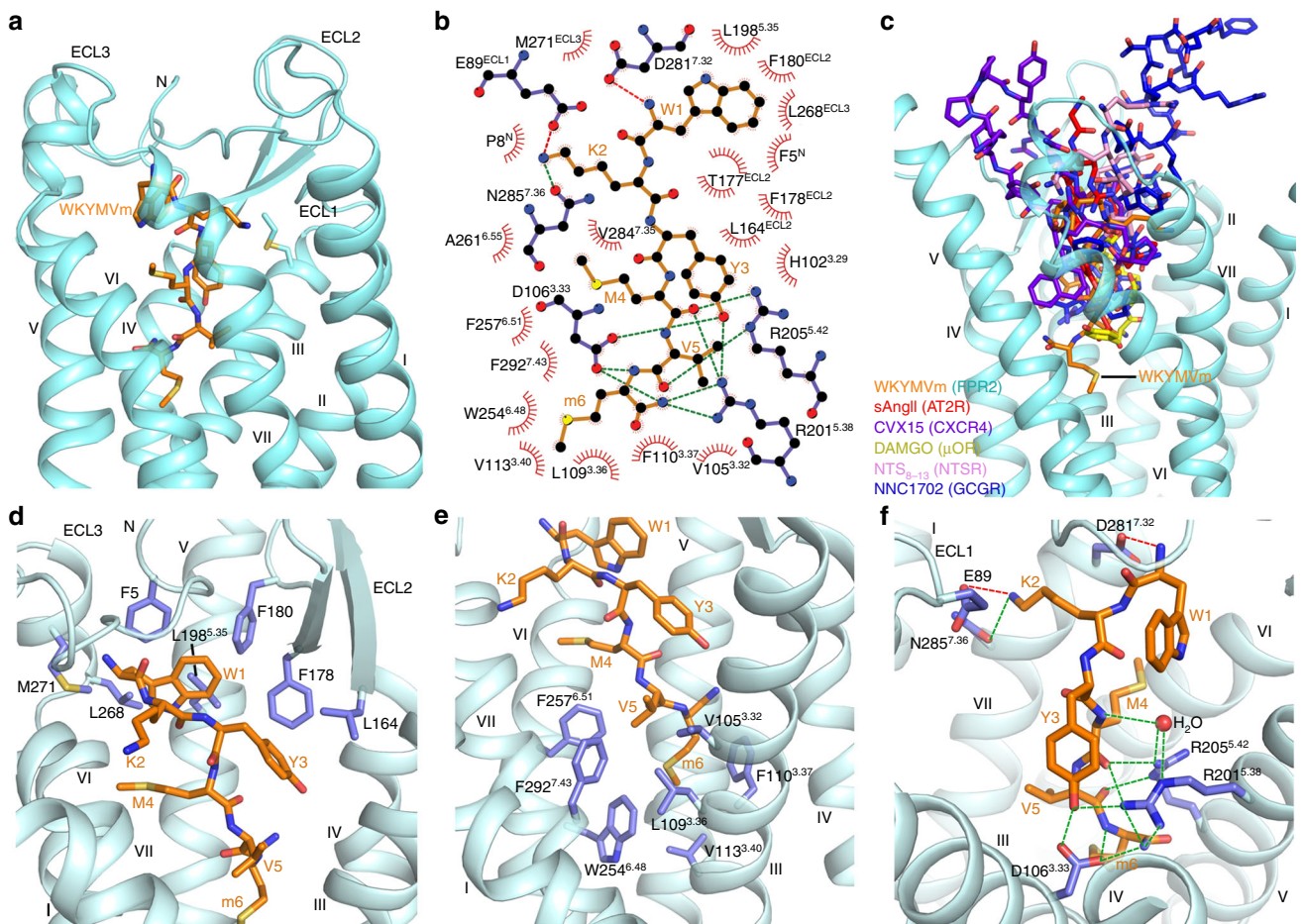

**Fig. 3 Ligand binding mode of FPR2 to WKYMVm. a** Binding pocket of WKYMVm in FPR2. The receptor in the FPR2-WKYMVm structure is shown in cyan cartoon representation. The ligand WKYMVm is shown as orange sticks. **b** Schematic representation of interactions between FPR2 and WKYMVm analyzed using the LigPlot[+] program[44]. Salt bridges and hydrogen bonds are shown as red and green dashed lines, respectively. The stick drawings of FPR2 and WKYMVm are colored blue and orange, respectively. **c** Comparison of the peptide ligand-binding sites in peptide-bound GPCR structures. WKYMVm in FPR2 (orange), sAngII in AT2R (red, PDB code: 5XJM), CVX15 in CXCR4 (purple, PDB code: 3OE0), DAMGO in μOR (yellow, PDB code: 6DDF), NTS$_{8-13}$ in NTSR (pink, PDB code: 4GRV), and NNC1702 in GCGR (blue, PDB code: 5YQZ) are shown as sticks. Only the receptor in the FPR2-WKYMVm structure is shown in cyan cartoon representation for calrification. **d** Hydrophobic cluster that forms interactions with W1 and Y3 of WKYMVm. The receptor residues and ligand are shown as blue and orange sticks, respectively. **e** Hydrophobic cluster that forms interactions with V5 and m6 of WKYMVm. **f** Polar interactions between FPR2 and WKYMVm. Salt bridges and hydrogen bonds are shown as red and green dashed lines, respectively.

impaired binding of WK(FITC)YMVm and the ability of WKYMVm in stimulating Gα$_{\Delta6qi4myr}$-mediated IP production (Fig. 4c, g and Supplementary Tables 1, 2). To further stabilize the binding of the peptide C terminus to the receptor, another polar residue R205$^{5.42}$ forms two hydrogen bonds with the main chain carbonyls of M4 and V5 (Fig. 3f). This was reflected by a complete loss of WK(FITC)YMVm binding and a significant reduction of the agonistic potency of WKYMVm for the mutant R205$^{5.42}$A (Fig. 4c, g and Supplementary Tables 1, 2). Moreover, a water molecule establishes a "bridge" between the main chain of the peptide residue M4 and the receptor residues R201$^{5.38}$ and R205$^{5.42}$, further stabilizing the peptide conformation and strengthening the receptor-peptide binding (Fig. 3f).

In addition to the above polar interactions involving the C terminus of WKYMVm, the receptor-ligand binding is facilitated by three polar interactions between the N terminus of the peptide and the extracellular region of helix VII and ECL1 in FPR2. The negatively charged residue D281$^{7.32}$ forms a salt bridge with the N-terminal NH$_2$- group of WKYMVm, while the only charged residue in the peptide, K2, is engaged in a salt bridge with the residue E89 in ECL1 and a hydrogen bond

with N285$^{7.36}$ in helix VII (Fig. 3f). However, in contrast to the significant impairment of WK(FITC)YMVm binding and IP production due to the D106$^{3.33}$A, R201$^{5.38}$A, and R205$^{5.42}$A substitutions, the mutants E89A/G, D281$^{7.32}$A, and N285$^{7.36}$A had much less impact on ligand recognition and receptor activation (Fig. 4d, h and Supplementary Tables 1, 2). These data suggest that these polar interactions with the N terminus of WKYMVm are less critical for recognition of the peptide ligand and its agonistic potency.

WKYMVM-NH$_2$, a derivative of WKYMVm with the substitution of L-methionine at the C terminus, is less effective on activating FPR2 than WKYMVm with an over 100-fold reduction in EC$_{50}$ in stimulating phosphoinositide hydrolysis[29]. Molecular docking of WKYMVM-NH$_2$ (Supplementary Data 1) revealed a rotation of the C-terminal amide relative to the binding pose of WKYMVm due to the alteration of chiral carbon. This movement breaks the hydrogen bonds between the C-terminal NH$_2$- group of the peptide and the receptor residues D106$^{3.33}$ and R201$^{5.38}$, and disrupts the polar interaction network established by the peptide C terminus (Supplementary Fig. 4), which is important for stabilizing receptor-peptide binding.

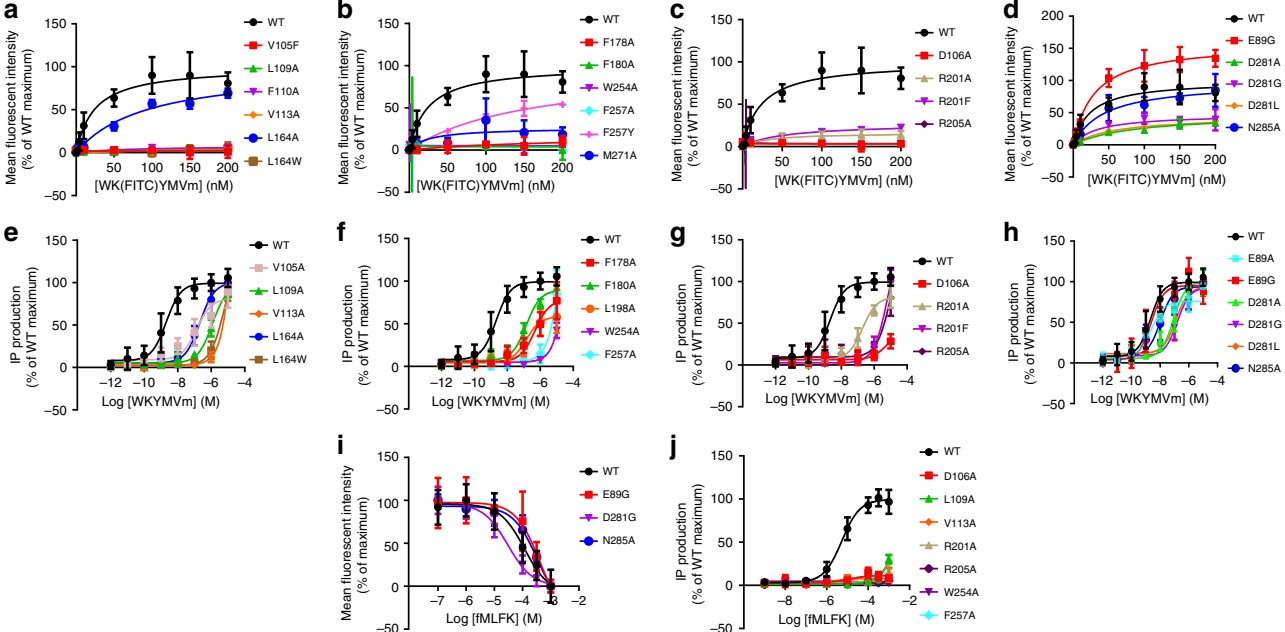

**Fig. 4 Ligand binding and signaling assays of the wild-type (WT) FPR2 and mutants.** Dose–response curves were generated from at least three independent experiments performed in triplicate. Data are shown as mean ± s.e.m. Source data are provided as a Source Data file. **a–d** Saturation binding of WK(FITC)YMVm. See Supplementary Table 1 for detailed statistical evaluation. **e–h** WKYMVm-induced IP accumulation assay using a chimeric Gα protein Gα$_{Δ6qi4myr}$. See Supplementary Table 2 for detailed statistical evaluation. **i** Binding of WK(FITC)YMVm inhibited by fMLFK. See Supplementary Table 1 for detailed statistical evaluation. **j** fMLFK-induced IP accumulation assay. See Supplementary Table 2 for detailed statistical evaluation.

**Molecular docking of formyl peptides**. The *E. coli*-derived chemotactic peptide fMLF is the shortest formyl peptide that exhibits full agonistic activity. However, it acts as a weak agonist for FPR2 with a 2000-fold lower potency in inducing Ca$^{2+}$ mobilization and an over 400-fold lower binding affinity compared to those for FPR1[4]. To investigate the binding mode of formyl peptide in FPR2, molecular docking of this tripeptide to the FPR2 crystal structure was performed (Supplementary Data 2). Similar to the C-terminal residue m6 of WKYMVm in the crystal structure, the formylated methionine at the N terminus of fMLF reaches deep into the ligand-binding pocket with the *N*-formyl group hydrogen bonding with the FPR2 residues D106$^{3.33}$ and R201$^{5.38}$ in the docking model (Fig. 5a), while the two hydrophobic residues L2 and F3 in fMLF occupy similar binding sites to those of V5 and Y3 in WKYMVm (Fig. 5b and Supplementary Fig. 5a, b). Furthermore, two basic residues in receptor helix V, R201$^{5.38}$ and R205$^{5.42}$, which provide the only positively charged binding interface in the ligand-binding cavity of FPR2, anchor the C-terminal COO- group of fMLF through two salt bridges (Fig. 5a).

To provide a structural basis for different behaviors of fMLF at FPR1 *vs.* FPR2, molecular docking of fMLF to a FPR2-based model of FPR1 was also carried out (Supplementary Data 3). Comparison of the FPR1-fMLF and FPR2-fMLF models revealed major difference in the interaction mode between the C-terminal COO− group of fMLF and the receptor. Unlike FPR2, in which the COO− group of the peptide forms salt bridges with R201$^{5.38}$ and R205$^{5.42}$, this C-terminal acidic group of fMLF may form ionic interactions with the residues R84$^{2.63}$ and K85$^{2.64}$ at the extracellular tip of helix II in FPR1 (Fig. 5c). These interactions are not possible in FPR2 as these two basic residues are replaced with non-charged residues, S84$^{2.63}$ and M85$^{2.64}$. The FPR1 residues R84$^{2.63}$ and K85$^{2.64}$ have been suggested to play critical roles in fMLF binding and receptor function of FPR1[32]. In contrast to the binding mode of fMLF in FPR2, where the residues D106$^{3.33}$, R201$^{5.38}$, and

R205$^{5.42}$ provide major polar contacts with the peptide, the extra ionic interactions between the peptide COO− group and R84$^{2.63}$ and K85$^{2.64}$ in FPR1 may contribute to the high binding affinity of fMLF at FPR1. This agrees with previous data showing that the FPR2 mutants S84$^{2.63}$R and M85$^{2.64}$K displayed greatly increased binding affinity for [$^3$H]fMLF[33]. These data support that these two residues in helix II are key factors governing fMLF recognition by FPR1, which align well with our molecular docking results.

It has been suggested that the length of the formyl peptide and its C-terminal charge are determinants for optimal agonistic activity at FPR2[4]. The tetrapeptide fMLFK and pentapeptide fMLFII showed increased binding affinity and agonist potency over fMLF at FPR2. It was also reported that the polar residue D281$^{7.32}$ was crucial for the interaction of FPR2 with certain formyl peptides. It was proposed that this negatively charged residue in FPR2 was repulsive with the C-terminal COO− group of fMLF and the negatively charged glutamate residue in fMLFE, but forms a stable interaction with the positively charged lysine in fMLFK[4]. In our docking models of the formyl peptides, the tetrapeptide fMLFK binds to FPR2 with its C-terminal residue K4 forming a salt bridge with either E89 or D281$^{7.32}$ (Fig. 5d, Supplementary Fig. 5c, and Supplementary Data 4). This interaction was verified by a competition binding assay of fMLFK with WK(FITC)YMVm, showing that the mutation E89G decreased the binding affinity for fMLFK by about 4-fold compared to the wild-type receptor while the mutant D281$^{7.32}$G displayed a slightly higher binding affinity to this tetrapeptide than the wild type (Fig. 4i and Supplementary Table 1). These data suggest that E89 plays a more important role in recognizing fMLFK and is likely the binding partner for the positively charged lysine at the C terminus of the peptide. In addition to this ionic interaction, K4 may also form a hydrogen bond with N285$^{7.36}$ (Fig. 5d). These polar interactions most likely contribute to the increased binding affinity and agonistic potency of fMLFK relative to fMLF. Likewise, molecular docking

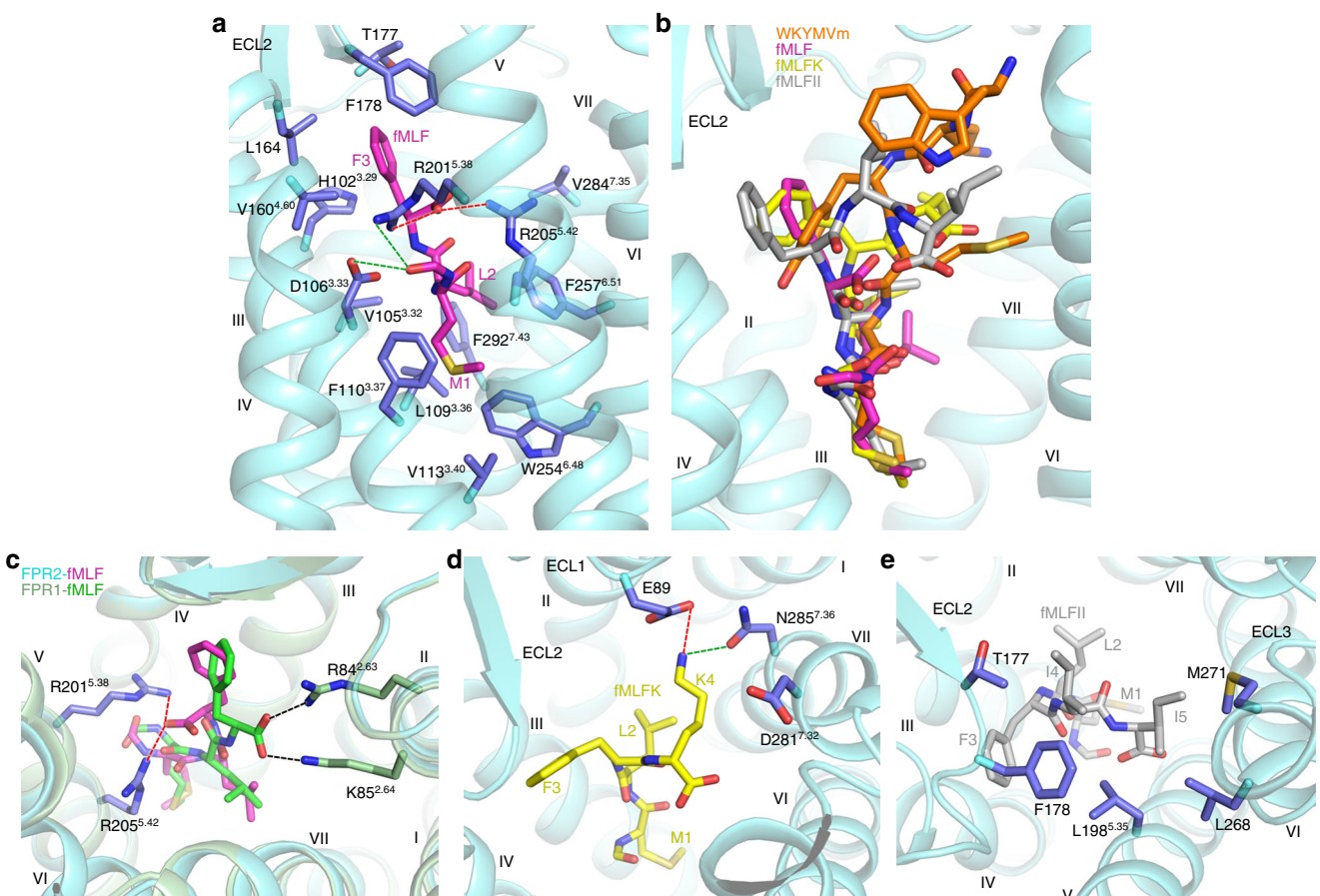

**Fig. 5 Molecular docking of formyl peptides. a** Docking model of FPR2-fMLF. The receptor is shown in cyan cartoon representation. The peptide fMLF and the FPR2 residues that may form interactions with fMLF are shown as sticks and colored magenta and blue, respectively. Salt bridges and hydrogen bonds are shown as red and green dashed lines, respectively. **b** Docking poses of fMLF, fMLFK, and fMLFII in comparison with the binding pose of WKYMVm in FPR2. The peptides WKYMVm, fMLF, fMLFK, and fMLFII are shown as orange, magenta, yellow, and gray sticks, respectively. **c** Comparison of the fMLF docking poses in FPR1 and FPR2. The model of FPR1-fMLF is colored light green (FPR1) and green (fMLF), while the model of FPR2-fMLF is in cyan (FPR2) and magenta (fMLF). The residues that may form salt-bridge interactions with the C-terminal COO− group of fMLF are shown as sticks and colored light green (R84[2.63] and K85[2.64] in FPR1) and blue (R201[5.38] and R205[5.42] in FPR2). The salt bridges are displayed as black and red dashed lines in FPR1 and FPR2, respectively. **d** Docking pose of fMLFK in FPR2. fMLFK is shown as yellow sticks. The receptor residues that may form polar interactions with the peptide residue K4 are shown as blue sticks. The polar interactions are shown as red (salt bridge) and green (hydrogen bond) dashed lines. **e** Docking pose of fMLFII in FPR2. fMLFII is shown as gray sticks. The receptor residues that may form hydrophobic interactions with the peptide residues I4 and I5 are shown as blue sticks.

identified extensive interactions between the two isoleucine residues at the C terminus of the pentapeptide fMLFII and the receptor ECL2, ECL3 and helices V and VI (Fig. 5e and Supplementary Data 5), a similar binding site to that of the residue W1 in WKYMVm (Fig. 5b and Supplementary Fig. 5a, d), which may lead to the improved activity of the pentapeptide. In contrast to the different behaviors of various formyl peptides at FPR2, peptide length and the composition of the peptide C terminus are not critical to FPR1 binding[4]. This may arise from a broader and less negative charged binding cavity on the extracellular side of the ligand-binding pocket in FPR1, which accommodates the peptide C terminus with fewer contacts and no preference for charges (Supplementary Fig. 6).

## Discussion

Among the FPR2 ligands with diverse structures, WKYMVm is by far the most potent peptide agonist for FPR2[2]. It exhibits stronger potency in activating FPR2 than FPR1 and FPR3, displaying a 40–300-fold higher $EC_{50}$ in mobilizing intracellular calcium (FPR2, 75 pM and FPR1, 25 nM[34]; FPR2, 75 pM and

FPR3, 3 nM[35]). Sequence alignment of the three FPRs reveals that most of the key residues involved in WKYMVm binding are conserved except for F5, E89, L164[4.64], L198[5.35], R201[5.38], and D281[7.32] (Supplementary Fig. 7), suggesting that these six residues may be determinants for binding selectivity of WKYMVm. In FPR1, the substitutions of F5 with serine and E89 with glycine may disrupt the interaction between the peptide and the extracellular loops of the receptor, leading to a decreased binding ability of the peptide. In FPR3, the bulky residue W[4.64] would most likely cause a spatial clash with the peptide residue Y3 to decrease the binding affinity, while the replacement of L[5.35] with A[5.35] may weaken the hydrophobic interaction with the peptide residue W1. This is supported by the ligand binding and IP accumulation assays, showing that the mutation L164[4.64]W severely impaired the WK(FITC)YMVm binding and receptor signaling, and the mutation L198[5.35]A reduced the agonist potency of WKYMVm by about 70-fold (Fig. 4a, e, f and Supplementary Tables 1, 2). Furthermore, instead of a polar arginine, the residue at position 5.38 in FPR3 is a hydrophobic phenylalanine, which disturbs the polar interaction network between the receptor helices III and V and the peptide, and probably mediates

selectivity. This agrees with the fact that the mutation R201$^{5.38}$F substantially reduced the WK(FITC)YMVm binding and WKYMVm-induced IP accumulation (Fig. 4c, g and Supplementary Tables 1, 2). Similarly, the FPR2 residue D$^{7.32}$ is replaced by glycine and leucine in FPR1 and FPR3, respectively, preventing the salt-bridge interaction with the N-terminal NH$_2$- group of WKYMVm. Indeed, the FPR2 mutants D281$^{7.32}$G and D281$^{7.32}$L exhibited decreased binding of WK(FITC)YMVm and an impaired ability to induce IP production (Fig. 4d, h and Supplementary Tables 1, 2). In addition, although the FPR2 residue F257$^{6.51}$ is substituted with a similar aromatic tyrosine residue in FPR1 and FPR3, the extra hydroxyl group may form a spatial clash with the peptide ligand, which is supported by a 7-fold reduction of binding affinity of WK(FITC)YMVm for the mutant F257$^{6.51}$Y (Fig. 4b and Supplementary Table 1). These insights gained from the FPR2-WKYMVm structure will facilitate the development of selective drug molecules by targeting the variable regions of the ligand-binding pocket.

It was reported that the related peptide MKYMPM-NH$_2$ was inactive when the residue M6 was either eliminated or replaced by glycine[29]. Furthermore, previous studies of D-type amino acid-containing peptide analogs of MKYMPM-NH$_2$ and WKYMVM-NH$_2$ revealed that none of the peptides with D-type amino acid substitutions was as effective as the original peptides, except for the ones with the D-Met6 substitution[29]. These data suggest that the peptide C terminus is a determinant of its biological activity. Indeed, among the alanine mutations of FPR2 tested in the G$\alpha_{\Delta6qi4myr}$-mediated IP accumulation assay, D106$^{3.33}$A, L109$^{3.36}$A, V113$^{3.40}$A, R205$^{5.42}$A, W254$^{6.48}$A, and F257$^{6.51}$A exhibited the largest effect on receptor signaling, showing a significantly impaired agonistic potency of WKYMVm (Fig. 4e–g and Supplementary Table 2). In the FPR2-WKYMVm structure, these six residues all locate at the bottom of the ligand-binding pocket and mainly form interactions with the C-terminal residue m6 of the peptide (Fig. 3e, f). The highly conserved class A GPCR residues I/V$^{3.40}$ and W$^{6.48}$ have been suggested to be involved in stimulating receptor activation through their conformational changes[16,22]. The interactions between the WKYMVm residue m6 and the FPR2 residues V113$^{3.40}$ and W254$^{6.48}$ are most likely crucial for triggering the conformational rearrangement of these conserved motifs to relay the agonist-induced conformational changes in the ligand-binding pocket to the cytoplasmic domain, while the other key residues within the sub-pocket may play a critical role in mediating receptor-ligand binding and/or stabilizing the receptor active conformation.

Although the formyl peptides and WKYMVm adopt different binding modes to FPR2, these peptide agonists may activate the receptor using a similar mechanism. This is supported by the fMLFK-induced IP accumulation assay, showing that the alanine replacements of L106$^{3.33}$, L109$^{3.36}$, V113$^{3.40}$, R201$^{5.38}$, R205$^{5.42}$, W254$^{6.48}$, and F257$^{6.51}$ abolished FPR2-mediated cell signaling (Fig. 4j and Supplementary Table 2). In the docking models of FPR2 bound to the formyl peptides, these residues all interact with the N-terminal formylated M1 of the peptides, suggesting that the N terminus of the formyl peptides activates the receptor in a similar manner to the C terminus of WKYMVm. These data indicate that the bottom region of the ligand-binding pocket in FPR2 plays an important role in regulating receptor activation, and can be considered as a drug target site for drug molecule design.

Collectively, the FPR2-WKYMVm complex structure provides molecular details regarding ligand recognition by FPR2 and other formyl peptide receptors. It is expected that understanding of the structural basis for FPR2 interaction with a variety of ligands will enable structure-based drug discovery targeting this physiologically important GPCR family.

## Methods

**Cloning and protein expression**. The gene of human FPR2 was codon-optimized and synthesized by Genewiz. It was cloned into a modified pTT5 vector (Invitrogen) containing an expression cassette with a hemagglutinin (HA) signal sequence followed by a Flag tag at the N terminus and a PreScission protease site and a 10 × His-tag at the C terminus. The N-terminal residues M1-E2 of FPR2 were replaced by the fusion partner bRIL using overlap extension PCR. Five residues (E347–M351) were truncated at the C terminus. A point mutation S211$^{5.48}$L was introduced into the FPR2 gene by standard QuikChange PCR. The codon-optimized DNA sequence and all primer sequences are shown in Supplementary Table 4.

HEK293F cells (Invitrogen) were grown in suspension with the starting density at $0.6 \times 10^6$ cells ml$^{-1}$ in 5% CO$_2$ at 37 °C. Once the cell density was increased to $1.2 \times 10^6$ cells ml$^{-1}$, the cells were transfected with the plasmid of FPR2 using the FreeStyle$^{TM}$ 293 Expression system (Invitrogen). Cells were collected after 48 h post transfection by centrifugation, and then stored at –80 °C until further use.

**Purification of the FPR2-WKYMVm complex**. Cells were disrupted by thawing the frozen cell pellets in a hypotonic buffer containing 10 mM HEPES, pH 7.5, 10 mM MgCl$_2$, 20 mM KCl, and EDTA-free protease inhibitor cocktail (Roche) with the ratio of 1 tablet per 100 ml buffer. Extensive washing of the cell membranes was performed by repeated centrifugation and dounce homogenization in the same hypotonic buffer. The cell debris was isolated by centrifugation at 160,000×$g$ for 30 min, and then resuspended in a high osmotic buffer containing 10 mM HEPES, pH 7.5, 1 M NaCl, 10 mM MgCl$_2$, and 20 mM KCl by dounce homogenization to remove soluble and membrane associated proteins. This step was repeated twice. The membranes were then washed by the hypotonic buffer to remove the high concentration of NaCl. The purified membranes were resuspended in 10 mM HEPES, pH 7.5, 30% (v/v) glycerol, 10 mM MgCl$_2$, 20 mM KCl, and EDTA-free complete protease inhibitor cocktail, flash-frozen with liquid nitrogen and stored at –80 °C until further use.

The purified membranes were thawed on ice in the presence of EDTA-free protease inhibitor cocktail, 100 μM WKYMVm and 2 mg ml$^{-1}$ iodoacetamide (Sigma), and incubated at 4 °C for 1 h. The membranes were then solubilized in 50 mM HEPES, pH 7.5, 150 mM NaCl, 0.5% (w/v) $n$-dodecyl-β-D-maltopyranoside (DDM, Anatrace), 0.1% (w/v) cholesterol hemisuccinate (CHS, Sigma), and 100 μM WKYMVm at 4 °C for 3 h. The supernatant was isolated by centrifugation at 160,000×$g$ for 30 min and incubated with TALON IMAC resin (Clontech) supplemented with 10 mM imidazole, pH 7.5 overnight at 4 °C. The resin was then washed with fifteen column volumes of washing buffer 1 containing 25 mM HEPES, pH 7.5, 150 mM NaCl, 10% (v/v) glycerol, 0.05% (w/v) DDM, 0.01% (w/v) CHS, 30 mM imidazole, and 100 μM WKYMVm followed by ten column volumes of washing buffer 2 that contains 25 mM HEPES, pH 7.5, 150 mM NaCl, 10% (v/v) glycerol, 0.05% (w/v) DDM, 0.01% (w/v) CHS, 5 mM ATP, and 100 μM WKYMVm. The receptor was then eluted with four column volumes of 25 mM HEPES, pH 7.5, 150 mM NaCl, 10% (v/v) glycerol, 0.05% (w/v) DDM, 0.01% (w/v) CHS, 300 mM imidazole, and 100 μM WKYMVm. PD MiniTrap G-25 column (GE Healthcare) was used to remove imidazole. The receptor was then treated overnight with His-tagged PreScission protease (custom-made) and His-tagged PNGase F (custom-made) to remove the C-terminal His-tag and deglycosylate the receptor. PreScission protease, PNGase F and the cleaved His-tag were removed by incubating the protein sample with Ni-NTA resin (Qiagen) at 4 °C for 1 h. The complex protein was then concentrated to 10–20 mg ml$^{-1}$ and analysed by SDS-PAGE and analytical size-exclusion chromatography for purity and homogeneity.

**Lipidic cubic phase crystallization**. The FPR2-WKYMVm sample was mixed with molten lipid (monoolein and cholesterol 9:1 by mass) at a weight ratio of 1:1.5 (protein:lipid) using two syringes to create a lipidic cubic phase (LCP). The mixture was dispensed onto glass sandwich plates (Shanghai FAstal BioTech) in 40 nl drop and overlaid with 800 nl precipitant solution using a Gryphon robot (Art-Robbins). Protein reconstitution in LCP and crystallization trials were performed at room temperature (19–22 °C). Plates were placed in an incubator (Rock Imager, Formulatrix) and imaged at 20 °C automatically following a schedule. Crystals of FPR2-WKYMVm complex appeared after 4 days and grew to full size ($50 \times 50 \times 5$ μm$^3$) within two weeks in 0.1 M Tris, pH 7.0–7.6, 30–36% (v/v) PEG500 DME, 2–5% PPG400, 50–150 mM CH$_3$COOLi, and 100 μM WKYMVm. The crystals were harvested directly from LCP using 30 and 50 μm micro mounts (M2-L19-30/50, MiTeGen), and flash frozen in liquid nitrogen.

**Diffraction data collection and structure determination**. X-ray diffraction data were collected at the SPring-8 beam line 41XU, Hyogo, Japan, using a EIGER16M detector (X-ray wavelength 1.0000 Å). The crystals were exposed with a 10 μm × 9 μm mini-beam for 0.2 s and 0.2° oscillation per frame. Most crystals diffracted to 2.4–3.5 Å resolution. XDS[36] was used to integrate and scale the data from 28 best-diffracting crystals. The initial phase was obtained by molecular replacement using Phaser[37] with the receptor portion of μOR (PDB accession code: 5C1M) and the structure of bRIL (PDB accession code: 1M6T) as search models. The MR solution contains one bRIL–FPR2 molecule in the asymmetric unit. Refinement was

performed using PHENIX[38] and BUSTER[39], and manual examination and rebuilding of the refined coordinates were carried out in COOT[40] using both $|2F_o|-|F_c|$ and $|F_o|-|F_c|$ maps. The Ramachandran plot analysis indicates that 100% of the residues are in favored (95.5%) or allowed (4.5%) regions (no outliers). The final model includes 320 residues (T3-E322) of FPR2 and residues A1-L106 of bRIL.

**Inositol phosphate accumulation assay.** Flag-tagged wild-type and mutant FPR2s were cloned into the expression vector pTT5 (Invitrogen) and expressed in HEK293 cells (Invitrogen) along with the chimeric Gα protein Gα$_{Δ6qi4myr}$ at the ratio of plasmids of 1:2 (w/w). Cells were routinely tested for mycoplasma contamination. Cells were harvested 48 h post transfection. Cell-surface expression of the receptor was measured by mixing 10 μl cells and 15 μl Monoclonal ANTI-FLAG M2-FITC antibody (Sigma, F4049; 1:100 diluted by TBS supplemented with 4% BSA). After 20 min, the fluorescence signal on the cell surface was measured by a FCM (flow cytometry) reader (Millipore).

IP1 accumulation was measured using an IP-One Gq assay kit (Cisbio Bioassays, 62IPAPEB) following the manufacturer's instruction. In brief, the cells were plated in 384-well plates (20,000 cells per well) and treated with different concentrations of WKYMVm (1 pM–10 μM) or fMLFK (1 nM–1 mM) diluted in stimulation buffer at 37 °C for 90 min. Then 3 μl cryptate-labeled anti-IP1 monoclonal antibody and 3 μl d2-labeled IP1, which were pre-diluted in Lysis Buffer (1:20), were added to the wells, and incubated at room temperature for 1 h. Plates were read in a Synergy$^{TM}$ H1 Operator (BioTek) with excitation at 330 nm and emission at 620 and 665 nm. The IP1 production was calculated according to a standard dose–response curve. Data were analyzed using Prism 7.0.

**Ligand binding assay.** Flag-tagged wild-type and mutant FPR2s were cloned into the pTT5 vector and expressed in HEK293F cells. The cells were harvested, and washed with Hanks' Balanced Salt Solution (HBSS) buffer supplemented with 0.5% bovine serum albumin and 20 mM HEPES, pH 7.4. The cells were then resuspended in the same buffer to the final concentration of $1 \times 10^6$ cells per ml. The cell-surface expression was measured as mentioned above. For saturation binding of WKYMVm, cells were plated in 96-well plates (100,000 cells per well) and incubated with increasing concentrations of fluorescein isothiocyanate (FITC)-conjugated peptide WK(FITC)YMVm (1 nM–200 nM) on ice for 1 h. Mean fluorescent intensity of each well was then read by a FCM (flow cytometry) reader (Millipore). Total binding and nonspecific binding were measured in the absence and presence of unlabeled ligand (200 μM WKYMVm), respectively. For competitive binding of fMLFK, cells were plated in 96-well plates (100,000 cells per well), and incubated with WK(FITC)YMVm at 4 °C for 1 h. The following concentrations of WK(FITC)YMVm were used: 10 nM (wild type, L81$^{2.60}$F, E89G, and V105$^{3.32}$A), 30 nM (D281$^{7.32}$G), 40 nM (H102$^{3.29}$F, V160$^{4.60}$A, L268$^{ECL3}$A, N285$^{7.36}$A, and F292$^{7.43}$A), and 100 nM (L164$^{4.64}$A and T177A). Then increasing concentrations of fMLFK (100 nM–1 mM) were added, and incubated for another 1 h on ice. Mean fluorescent intensity values were measured by flow cytometry. Data were analyzed using Prism 7.0.

**Molecular docking of peptide ligands.** The structure of FPR1 was modeled using the crystal structure of FPR2-WKYMVm as a template and refined with the Advanced Homology Modeling and Minimization implemented in the Schrödinger suite[41]. The modeled structure of FPR1 and the crystal structure of FPR2-WKYMVm were prepared using the Protein Preparation Wizard implemented in the Schrödinger suite to add hydrogen atoms and the missing side chains of residues. The orientation of polar hydrogens and the protonated states of the receptor were then optimized. The overall structures were refined using OPLS3 forced field[42] with harmonic restraints on heavy atoms. The 3D structures of fMLF, fMLFK, fMLFII, and WKYMVM-NH$_2$ were generated and optimized using the Ligprep tool of the Schrödinger suite, and dockings of these ligands to FPR1 or FPR2 were performed with the Induced Fit Docking (IFD) tool as previously described[43]. IFD allows conformational changes of the receptor ligand-binding site upon docking of different ligands. The docking grid was centered on the centroid of WKYMVm. The docking simulations were performed with default settings except that the extra precision (XP) mode was used in the last docking round. The final pose of every docked peptide was selected from the top-scoring conformations using the binding mode of WKYMVm in the crystal structure as a reference. To verify the above method, we also applied the same docking algorithm on WKYMVm, showing a docking pose similar to the binding pose observed in the crystal structure (Cα r.m.s.d., 0.6 Å; all atom r.m.s.d., 2.2 Å).

**Reporting summary.** Further information on research design is available in the Nature Research Reporting Summary linked to this article.

## Data availability
Atomic coordinates and structure factors of the FPR2-WKYMVm structure have been deposited in the Protein Data Bank with accession code 6LW5. The source data underlying Fig. 4 and Supplementary Tables 1, 2 are provided as a Source Data file. Other data are available from the corresponding authors upon reasonable request.

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

## Acknowledgements

This work was supported by the National Key R&D Programs of China 2018YFA0507000 (Q.Z. and B.W.) and 2016YFA0502301 (Y.X.), the National Science Foundation of China grants 31825010, 31730027 (B.W.) and 81525024 (Q.Z.), CAS Strategic Priority Research Program XDB37000000 (B.W.), the Key Research Program of Frontier Sciences, CAS grants QYZDB-SSW-SMC024 (B.W.) and QYZDB-SSW-SMC054 (Q.Z.), and the National Science & Technology Major Project—Key New Drug Creation and Manufacturing Program, China grant 2018ZX09711002 (L.M.). The synchrotron radiation experiments were performed at the BL41XU of SPring-8 with approval of the Japan Synchrotron Radiation Research Institute (Proposal no. 2018A2527, 2018A2531, 2018B2527, 2018B2531, 2019A2543, and 2019B2543). We thank the beamline staff members K. Hasegawa, N. Mizuno, T. Kawamura, and H. Murakami of the BL41XU for help with X-ray data collection.

## Author contributions

T.C. optimized the construct, purified the FPR2 protein, performed crystallization trials, solved the structure, performed ligand binding and signaling assays, and helped with manuscript preparation. M.X. performed molecular docking. X.Z. helped with protein sample optimization and functional assays. Y.G. helped with functional assays. H.Z. and M.W. collected X-ray diffraction data. G.W.H. helped with structure determination. C.Y. and L.M. expressed the protein. R.D.Y. helped with data analysis and interpretation, and edited the manuscript. Y.X. oversaw the molecular docking and edited the manuscript. B.W. and Q.Z. initiated the project, planned and analysed experiments, supervised the research, and wrote the manuscript with input from all co-authors.

## Competing interests

The authors declare no competing interests.
