## [Peer Review File · Nature Communications]

Reviewers' comments:

Reviewer #1 (Remarks to the Author):

In this manuscript, Chen et al present a structural basis for different ligands binding to the formyl peptide receptor 2 (FPR2), a medically relevant GPCR. The authors determine an X-ray crystal structure of the FPR2 bound to the peptide agonist WKYMVm and compliment this with extensive mutagenesis, ligand binding, and cell signalling assays (IP1). In addition, to understand how other formyl peptides and lipoxin A4 bind to FPR1/2, the authors performed homology modelling and induced fit docking experiments with binding / functional assays. Overall, this work is of high quality, well written, and novel. The findings from this work provide important insight into how different ligands can selectivity bind to different formyl peptide receptors paving the way for future structure-based drug discovery.

As presented, I don't have any major concerns that would prevent publication, and I offer a few comments / suggestions that could be used to improve the manuscript.

- 1) A single mutation S211L was made to improve protein quality. There is no description on how or why this mutation was designed. This should be clarified, was it computationally predicted, screened for, or based on previous work?
- 2) The abstract suggests that docking and functional assays revealed key features about peptide binding and LXA4. However, functional assays were only performed for fMLFK, this could be better clarified in the abstract.
- 3) The docking of different ligands into FPR2 is a major feature of this work (Fig.5), it would be useful if these models were made available by either adding as supplemental information online or making available on request.

There is a typo bottom of page 11, 'C15-hydorxyl'

Reviewer #2 (Remarks to the Author):

Chen et al describe the three-dimensional structure of the human GPCR formyl peptide receptor 2 (FPR2) in complex with a potent peptide agonist WKYMVm. FPR2 is an inflammatory receptor that is promiscuous in its binding of multiple ligands including bacterial formyl peptides, lipoxin A4 mediators and other small molecules, and proteins. The manuscript includes docking of FPR2 with a number of other ligands (formyl peptides, the tetrapeptide fMLFK, LXA4) which are also used for competition binding and signaling studies against wild-type and structure-based mutants. Other insights include the comparison to other structures of GPCR-complexes and sequence homology to FPR1 and FPR3. The comparison to other GPCR-agonists complexes show the expected movements of transmembrane (TM) helices are consistent with agonism. The comparison to other peptide agonists/antagonists in their respective TM cavity indicates WKYMVm penetrates the cavity the deepest thus far. Sequence alignment of receptors FPR1-3 and the structural model of FPR1 based on the experimental model of FPR2 show differences in the electrostatic potential to show specificity of ligand binding.

The receptor complex provides insights into how its activation leads to inflammation. These include the N-terminal region and extracellular loops ECL1 and ECL2 forming a lid on top of WKYMVm. Are there specific contacts of the FPR2 N-terminus with WKYMVm? Fig. 1b suggests there are contacts, but it is difficult to determine from the figure (although none are shown in Fig. 3f). The fusion of BRIL to the N-terminus adds to this problem, as there is no discussion regarding crystal contacts that could influence the conformation of the N-terminus shown in Fig. 1B. The authors should address

these questions and might even be interested in making mutants at the FPR2 N-terminal region (aside from the reported F5A) to determine if it has a role in affinity or decreasing the koff rate of the peptide agonist. These should be addressed experimentally and/or discussed.

Over 30 mutants were described and used to study binding and/or signaling of WKYMVm and of docked fMLFK. Multiple mutants were made for the same residue (Ala, Gly, and others to test specific interactions). For about 50% of these mutants, a robust dose-response curve could not be determined. For those with a dose-response curve, range of Kds is 24-322 (Suppl table 2) and EC50s range from 1-600nM (Suppl table 2). The authors should acknowledge GPCR are dynamic proteins and many of the reported mutants might increase the dynamic nature of the receptor and affect binding and signaling. It is worthwhile for the authors to re-analyze their structures and mutants to determine if they can say anything about the dynamics or conformational changes regarding the results of the mutants they made.

The authors use the crystal structures as an opportunity to dock other ligands with the their experimentally determined structure of FPR2. With the exception of LXA4, the docking of formyl peptides with mutagenesis experiments produce insightful results for fMLFK. For LXA4, more work is necessary to justify the docking pose. Otherwise, it is my opinion that it should be omitted.

Some minor issues include:

On p. 9, the authors write "... C-terminal acidic group of fMLF tends to form ionic interactions ...". The word "tends" should be defined in this context or change "may".

How was the omit map calculated to remove any bias for the agonist in SFig. 1?

On page 9 of the Supplement bSample size; the number of independent experiments performed in technical triplicate. I think "technical" was inadvertently added.

Reviewer #3 (Remarks to the Author):

The authors provide a very interesting structural study on the formyl peptide receptor 2 (FPR2). With the first structure of a formyl peptide receptor (one of three isoforms) the authors open up structural insight on how the receptors interact with various ligands. The work is based on the active conformation structure of FPR2 receptor (as N-terminal BRIL fusion protein) in complex with peptide agonist WKYMVm. Using this structure as starting point ligand docking, mutagenesis studies, and homology modeling provided first structural ideas about how FPR2 and FPR1 have different specificities for various ligands including the eicosanoid lipoxin A4 (LXA4) The work is technically sound, clearly written with a good discussion, and is a nice fit for Nature Communications.

The authors may address a few points:

- When the authors describe the features of active GPCR conformations they might also cite the original work on the rhodopsin system.
- The authors use ligand docking extensively to distinguish the binding and effect of various ligands. The authors should comment whether they tried co-crystallization with other ligands and whether they were successful. Did the authors test their docking algorithm by using the WKYMVm ligand?
- The explanations are based on structures/ docking results. The authors might consider dynamic aspects of the receptor that can contribute to the functional outputs of the different receptor-

ligand complexes.

- The differences observed by LXA4 compared to peptide ligands are interesting. Did the authors try to crystallize the FPR2-LXA4 complex or to analyze the complex by other biophysical means?

- Figures. It might be helpful to show in some figures ECL1, ECL2, and ECL3 in different colors. Fig. 3c and 5b look overloaded. Perhaps they could be shown in addition as individual structures in the supplement.

Responses to the reviewers' comments

Reviewer #1 (Remarks to the Author):

In this manuscript, Chen et al present a structural basis for different ligands binding to the formyl peptide receptor 2 (FPR2), a medically relevant GPCR. The authors determine an X-ray crystal structure of the FPR2 bound to the peptide agonist WKYVM and compliment this with extensive mutagenesis, ligand binding, and cell signalling assays (IP1). In addition, to understand how other formyl peptides and lipoxin A4 bind to FPR1/2, the authors performed homology modelling and induced fit docking experiments with binding / functional assays. Overall, this work is of high quality, well written, and novel. The findings from this work provide important insight into how different ligands can selectively bind to different formyl peptide receptors paving the way for future structure-based drug discovery.

As presented, I don't have any major concerns that would prevent publication, and I offer a few comments / suggestions that could be used to improve the manuscript.

1) A single mutation S211L was made to improve protein quality. There is no description on how or why this mutation was designed. This should be clarified, was it computationally predicted, screened for, or based on previous work?

— We thank the reviewer for this comment. The mutation S211L was designed by comparing the amino acid sequences of FPR2 and some other class A GPCRs that share high sequence similarity with FPR2 (35-45%) and have structures determined, including CXCR4, CCR5, C5 α , μ OR and DP₂. Sequence alignment of these receptors revealed that the residue at position 5.48 (residue numbering using the Ballesteros Weinstein nomenclature) in FPR2 is hydrophilic (serine), while the corresponding residue in the other receptors is hydrophobic (leucine, isoleucine or valine) (see right figure). This hydrophobic residue, which locates on the external surface of the receptor, may stabilize the conformation of the receptor helical bundle by forming a hydrophobic interaction network with its neighboring residues. Thus, the mutation S211^{5.48}L was introduced in FPR2 in order to improve protein stability. As expected, this mutation greatly improved protein homogeneity and yield (see right figure), and thus facilitated structure determination.

a, Sequence alignment of FPR2, DP₂, C5 α , CXCR4, CCR5 and μ OR. The residues at position 5.48 are highlighted in a green box. **b**, Analytical size-exclusion chromatography of the FPR2 mutant S211^{5.48}L. The traces of the mutant and the protein without any mutation are colored red and black, respectively. The data show that the S211^{5.48}L mutant has higher protein yield and higher monomer:aggregation ratio, which indicates better protein homogeneity.

To make the above clear in the manuscript, the statement “It was designed by switching the hydrophilic residue to a hydrophobic counterpart presented in several class A GPCRs with known structures, including chemokine receptors CXCR4 and CCR5, C5 α receptor, μ -opioid receptor and prostanoid receptor DP₂, which share high sequence similarity with FPR2 (35-

45%)¹³⁻¹⁷. This mutation may introduce extra hydrophobic interactions with its neighboring residues on the external surface of the receptor, aiming for better protein stability” has been added to paragraph 1, page 4 in the revised version.

2) *The abstract suggests that docking and functional assays revealed key features about peptide binding and LXA4. However, functional assays were only performed for fMLFK, this could be better clarified in the abstract.*

— We appreciate the reviewer’s comment. The result and discussion about LXA4 have been removed from the abstract and main text as Reviewer #2 suggested. Please see the response to the third comment of Reviewer #2.

3) *The docking of different ligands into FPR2 is a major feature of this work (Fig.5), it would be useful if these models were made available by either adding as supplemental information online or making available on request.*

— We followed the reviewer’s suggestion and have provided the models of FPR2-WKYMVM, FPR2-fMLF, FPR1-fMLF, FPR2-fMLFK and FPR2-fMLFII as Supplementary Data 1-5.

4) *There is a typo bottom of page 11, ‘C15-hydorxyl’*

— We thank the reviewer for this comment. The “molecular docking of the eicosanoid ligand LXA4” section has been removed as mentioned above.

Reviewer #2 (Remarks to the Author):

Chen et al describe the three-dimensional structure of the human GPCR formyl peptide receptor 2 (FPR2) in complex with a potent peptide agonist WKYMVm. FPR2 is an inflammatory receptor that is promiscuous in its binding of multiple ligands including bacterial formyl peptides, lipoxin A4 mediators and other small molecules, and proteins. The manuscript includes docking of FPR2 with a number of other ligands (formyl peptides, the tetrapeptide fMLFK, LXA4) which are also used for competition binding and signaling studies against wild-type and structure-based mutants. Other insights include the comparison to other structures of GPCR-complexes and sequence homology to FPR1 and FPR3. The comparison to other GPCR-agonists complexes show the expected movements of transmembrane (TM) helices are consistent with agonism. The comparison to other peptide agonists/antagonists in their respective TM cavity indicates WKYMVm penetrates the cavity the deepest thus far. Sequence alignment of receptors FPR1-3 and the structural model of FPR1 based on the experimental model of FPR2 show differences in the electrostatic potential to show specificity of ligand binding.

1) *The receptor complex provides insights into how its activation leads to inflammation. These include the N-terminal region and extracellular loops ECL1 and ECL2 forming a lid on top of WKYMVm. Are there specific contacts of the FPR2 N-terminus with WKYMVm? Fig. 1b suggests there are contacts, but it is difficult determine from the figure (although none are shown in Fig. 3f). The fusion of BRIL to the N-terminus adds to this problem, as there is no discussion regarding crystal contacts that could influence the conformation of the N-terminus shown in Fig. 1B. The authors should address these questions and might even be interested in making mutants at the FPR2 N-terminal region (aside from the reported F5A) to determine if it has a role in affinity or decreasing the koff rate of the peptide agonist. These should be addressed experimentally and/or discussed.*

— We appreciate the reviewer’s comment. In the FPR2-WKYMVm crystal structure, the receptor N terminus forms limited hydrophobic contacts with the peptide agonist (Fig. 3d). However, as the reviewer pointed out, the N-terminal bRIL fusion protein, which plays a

critical role in mediating crystal packing (see figure below), may alter the conformation of the flexible N terminus. Thus, the contact between the receptor N terminus and the peptide may not be physiological relevant. Our mutagenesis studies show that the mutation F5A has little

Crystal packing of FPR2-WKYMVm complex. FPR2 is shown in cartoon representation and colored cyan. The bRIL fusion protein is shown in grey cartoon representation. WKYMVm is displayed as orange spheres.

effect on both WK(FITC)YMVm binding and receptor signaling (Supplementary Table 1 and Supplementary Table 2), suggesting that the interaction between the receptor N terminus and the peptide agonist is either not important for ligand recognition or introduced by crystal packing.

As the reviewer suggested, we have added the discussion regarding crystal contacts that may influence the conformation of the receptor N terminus to paragraph 1, page 5 as “However, the structure does not rule out the possibility that the conformation of the receptor N terminus was affected by the N-terminal bRIL fusion protein, which is involved in mediating crystal packing (Supplementary Fig. 1)”. A figure of crystal packing has also been included as Supplementary Fig. 1 in the revised version.

Furthermore, to make the potential effect of crystal contact on the interaction between the receptor N terminus and WKYMVm clear in the manuscript, the statement “In contrast to the substantial effect of most of the residues within the two hydrophobic clusters, the alanine mutation of the N-terminal residue F5 displays little effect in both assays (Supplementary Table 1 and Supplementary Table 2), suggesting that the interaction between the receptor N terminus and the peptide agonist is either not important for ligand recognition or introduced by crystal packing (Supplementary Fig. 1)” has been added to paragraph 1, page 7.

2) Over 30 mutants were described and used to study binding and/or signaling of WKYMVm and of docked fMLFK. Multiple mutants were made for the same residue (Ala, Gly, and others to test specific interactions). For about 50% of these mutants, a robust dose-response curve could not be determined. For those with a dose-response curve, range of Kds is 24-322 (Suppl table 2) and EC50s range from 1-600nM (Suppl table 2). The authors should acknowledge GPCR are dynamic proteins and many of the reported mutants might increase the dynamic nature of the receptor and affect binding and signaling. It is worthwhile for the authors to re-analyze their structures and mutants to determine if they can say anything about the dynamics or conformational changes regarding the results of the mutants they made.

— We thank the reviewer for this comment. Indeed, GPCRs are highly dynamic. Mutating residues in the ligand-binding pocket of FPR2, especially the hydrophobic residues within the two hydrophobic clusters that form extensive interactions with the residues W1, Y3, V5 and m6 of WKYMVm, may decrease the conformational stability of the ligand-binding pocket. These mutations may disturb the interactions between the residues within the hydrophobic clusters or disrupt the contacts of these residues with their neighboring residues to increase dynamics of the receptor helical bundle. Furthermore, the mutations of some residues such as V113^{3,40} and W254^{6,48}, which play critical roles in relaying the agonist-induced conformational changes in the ligand-binding pocket to the cytoplasmic domain, may impair the global conformational rearrangement that is required for receptor activation. All these above likely

contribute to the effect of mutations on ligand binding and receptor signaling in addition to the direct disruption of receptor-ligand interaction.

To make the above clear, the statement “The effect of these mutations on ligand binding and receptor signaling could be explained by direct disruption of the receptor-ligand interaction, destabilization of ligand-binding pocket conformation, and/or impairment of global conformational rearrangement required for receptor activation” has been added to the discussion of hydrophobic interactions between FPR2 and WKYMVm (paragraph 1, page 7).

3) *The authors use the crystal structures as an opportunity to dock other ligands with the their experimentally determined structure of FPR2. With the exception of LXA4, the docking of formyl peptides with mutagenesis experiments produce insightful results for fMLFK. For LXA4, more work is necessary to justify the docking pose. Otherwise, it is my opinion that it should be omitted.*

— We appreciated the reviewer’s comment. Functional assays for LXA4 are challenging due to the poor ability of this ligand to trigger calcium mobilization and IP accumulation. To verify the predicted binding mode of LXA4 in FPR2, we have performed additional assays by measuring the reduction in intracellular cAMP induced by WKYMVm or LXA4. The results show that both ligands induced measurable changes in the intracellular cAMP at the wild-type FPR2 (see figure below). The alanine mutation of V113^{3,40}, which interacts with WKYMVm but not LXA4, impaired the ability of WKYMVm to inhibit cAMP accumulation by about 400-fold while having minimal effect on the LXA4-induced cAMP reduction. These data suggest different binding modes of LXA4 and WKYMVm in FPR2 and align well with our docking model of LXA4. However, the mechanism underlying the ability of LXA4 to inhibit cAMP accumulation but not trigger calcium mobilization remains unknown and is outside the scope of this study. To avoid over speculation, the result and discussion about LXA4 have been omitted as suggested.

cAMP inhibition assay for FPR2. **a**, Wild-type (WT) FPR2; **b**, FPR2 mutant V113^{3,40}A. W-pep, WKYMVm. The indicated FPR2 constructs were transiently expressed in HEK293 cells for 24 h. The cells were plated into a 384-well plate, and stimulated for 30 min with different concentrations of WKYMVm or LXA4 together with 5 μ M forskolin. Accumulated cAMP was detected and the concentrations were shown.

Some minor issues include:

4) *On p. 9, the authors write “... C-terminal acidic group of fMLF tends to form ionic interactions ...”. The word “tends” should be defined in this context or change “may”.*

— The words “tends to” have been changed to “may” as suggested.

5) *How was the omit map calculated to remove any bias for the agonist in SFig. 1?*

— During structure refinement, the agonist was not modeled until the final stage of refinement with strong and unambiguous electron densities presented for all the residues of the peptide to avoid any bias. The $|F_o| - |F_c|$ omit map was calculated by directly removing the agonist from the model. To further confirm, we also calculated the composite omit map, which shows similar strong electron densities for the peptide (see figure below).

Electron densities of WKYMVm. **a**, Electron densities are contoured at 2.5σ from a $|F_o| - |F_c|$ omit map and colored blue. The receptor is shown in cyan cartoon representation. The peptide WKYMVm is shown as orange sticks. The disulfide bond is shown as yellow sticks. **b**, Electron densities are contoured at 2.0σ from a composite omit map ($2|F_o| - |F_c|$) and colored magenta.

6) On page 9 of the Supplement bSample size; the number of independent experiments performed in technical triplicate. I think “technical” was inadvertently added.

— The word “technical” in the foot notes of Supplementary Table 1 and 2 has been removed.

Reviewer #3 (Remarks to the Author):

The authors provide a very interesting structural study on the formyl peptide receptor 2 (FPR2). With the first structure of a formyl peptide receptor (one of three isoforms) the authors open up structural insight on how the receptors interact with various ligands. The work is based on the active conformation structure of FPR2 receptor (as N-terminal BRIL fusion protein) in complex with peptide agonist WKYMVm. Using this structure as starting point ligand docking, mutagenesis studies, and homology modeling provided first structural ideas about how FPR2 and FPR1 have different specificities for various ligands including the eicosanoid lipoxin A4 (LXA4) The work is technically sound, clearly written with a good discussion, and is a nice fit for Nature Communications.

The authors may address a few points:

1) When the authors describe the features of active GPCR conformations they might also cite the original work on the rhodopsin system.

— Per suggestion above, we have cited two references (references 20 and 26) about the conformation of the ‘ionic lock’ in both inactive and active structures of rhodopsin (Palczewski *et al.*, *Science* 289:739-745, 2000; Choe *et al.*, *Nature* 471:651-655, 2011) in discussion of the active conformation of the FPR2-WKYMVm structure (paragraph 2, page 5).

2) The authors use ligand docking extensively to distinguish the binding and effect of various ligands. The authors should comment whether they tried co-crystallization with other ligands and whether they were successful. Did the authors test their docking algorithm by using the WKYMVm ligand?

— Thank the reviewer for this comment. We did try to co-crystallize FPR2 with several formyl peptides, including fMLFK, fMLFIK and fMLFII, but failed. This most likely correlates with the poor quality of the FPR2-formyl peptide complexes (see figure below).

To test the docking algorithm, we docked the WKYMVm ligand in the crystal structure. The docking pose aligns well with the binding pose of the peptide in the crystal structure ($C\alpha$ r.m.s.d., 0.6 Å; all atom r.m.s.d., 2.2 Å) (see figure below). To make this clear in the manuscript, we have added the statement “To verify the above method, we also applied the same docking algorithm on WKYMVm, showing a docking pose similar to the binding pose observed in the crystal structure ($C\alpha$ r.m.s.d., 0.6 Å; all atom r.m.s.d., 2.2 Å)” to the Methods (paragraph 2, page 20).

a and **b**, Stability assays of FPR2-formyl peptide complexes. **a**, Analytical size-exclusive chromatography of FPR2 in apo state or bound to WKYMVm, fMLFK, fMLFIK or fMLFII. The black dashed lines indicate the peaks of protein aggregation and monomer, respectively. The data show that the formyl peptide-bound FPR2 complexes have lower protein yield and lower monomer:aggregation ratio compared to the FPR2-WKYMVm complex, which indicates poor protein homogeneity of the FPR2-formyl peptide complexes. **b**, Thermo-stability assay of FPR2 in apo state or bound to WKYMVm, fMLFK, fMLFIK or fMLFII. The green dashed line indicates the melting temperature (T_m) of the apo protein and the complexes of FPR2-formyl peptide. The red dashed line indicates the T_m of the FPR2-WKYMVm complex. The data show that the formyl peptide-bound FPR2 complexes have lower T_m than the FPR2-WKYMVm complex, indicating poor thermal stability of the FPR2-formyl peptide complexes. **c**, Comparison of the binding pose of WKYMVm in the crystal structure (orange) and the docking pose (cyan).

3) *The explanations are based on structures/docking results. The authors might consider dynamic aspects of the receptor that can contribute to the functional outputs of the different receptor-ligand complexes.*

— We appreciate the reviewer for this comment. The dynamic aspects of the receptor upon the binding of different ligands were considered in our molecular docking studies. The docking of ligands to FPR1 or FPR2 was performed using the Induced Fit Docking (IFD) tool, which allows side-chain conformational changes of the residues at the ligand-binding site upon docking of different ligands.

4) *The differences observed by LXA4 compared to peptide ligands are interesting. Did the authors try to crystallize the FPR2-LXA4 complex or to analyze the complex by other biophysical means?*

— Thank the reviewer for this comment. We did try to crystallize the FPR2-LXA4 complex but failed to get crystals. Nevertheless, we have performed additional functional assays to confirm the interaction of FPR2 with LXA4 and verify the predicted binding mode of LXA4 (please see the response to the third comment of Reviewer #2 for details). The data suggest different binding modes of LXA4 and WKYMVm in FPR2 and align well with our docking model of LXA4. However, the mechanism underlying the ability of LXA4 to inhibit cAMP

accumulation but not trigger calcium mobilization remains unknown and is outside the scope of this study. To avoid over speculation, the result and discussion about this ligand have been omitted as Reviewer #2 suggested.

5) Figures. It might be helpful to show in some figures ECL1, ECL2, and ECL3 in different colors.

— We followed the reviewer's suggestion and have revised Fig. 1 by highlighting the N terminus and extracellular loops of the receptor with different colors.

6) Fig. 3c and 5b look overloaded. Perhaps they could be shown in addition as individual structures in the supplement.

— The advice is well taken. The structure of FPR2-WKYMVm and some other peptide-bound GPCR structures shown in Fig. 3c are displayed as individual structures in Supplementary Fig. 3. The docking models of formyl peptides in Fig. 5b have been shown as Supplementary Fig. 5.